# Molecular Diagnosis of Koala Retrovirus (KoRV) in South Australian Koalas (*Phascolarctos cinereus*)

**DOI:** 10.3390/ani11051477

**Published:** 2021-05-20

**Authors:** Tamsyn Stephenson, Natasha Speight, Wai Yee Low, Lucy Woolford, Rick Tearle, Farhid Hemmatzadeh

**Affiliations:** 1School of Animal and Veterinary Sciences, University of Adelaide, Roseworthy 5371, Australia; natasha.speight@adelaide.edu.au (N.S.); lucy.woolford@adelaide.edu.au (L.W.); farhid.hemmatzadeh@adelaide.edu.au (F.H.); 2The Davies Livestock Research Centre, School of Animal and Veterinary Sciences, University of Adelaide, Roseworthy 5371, Australia; wai.low@adelaide.edu.au (W.Y.L.); rick.tearle@adelaide.edu.au (R.T.); 3Veterinary Diagnostics Laboratory, School of Animal and Veterinary Sciences, University of Adelaide, Roseworthy 5371, Australia

**Keywords:** koala retrovirus, KoRV, koala, Phascolarctos cinereus, diagnosis, PCR, qPCR, South Australia

## Abstract

**Simple Summary:**

Koala retrovirus (KoRV) is a significant threat to koalas across Australia. Koalas in northern koala populations (from New South Wales and Queensland) have KoRV inserted into their DNA and inherited to their offspring. Southern koala populations (from Victoria and South Australia) have KoRV infection spread through close contact of koalas. As such, there are koalas within South Australia that are not infected with KoRV. Accurate diagnosis of the infection of each koala is therefore fundamental for disease studies. Previous studies have shown differences in prevalence of different KoRV genes in the Mount Lofty Ranges Koala population; therefore, clarification is necessary. This study uses a large cohort (n = 216) and defines the diagnostic regions of the KoRV genome within the South Australian population. Using multiple molecular techniques, it demonstrates strong evidence for two clear groupings of koalas: KoRV positive and KoRV negative. Within this study, a population of 41% were shown to be KoRV positive and 57% were KoRV negative, with 2% inconclusive. This differentiation is of great importance when examining the clinical importance of KoRV infection within southern koalas.

**Abstract:**

Koala retrovirus, a recent discovery in Australian koalas, is endogenised in 100% of northern koalas but has lower prevalence in southern populations, with lower proviral and viral loads, and an undetermined level of endogenisation. KoRV has been associated with lymphoid neoplasia, e.g., lymphoma. Recent studies have revealed high complexity in southern koala retroviral infections, with a need to clarify what constitutes positive and negative cases. This study aimed to define KoRV infection status in Mount Lofty Ranges koalas in South Australia using RNA-seq and proviral analysis (n = 216). The basis for positivity of KoRV was deemed the presence of central regions of the KoRV genome (*gag* 2, *pol*, *env* 1, and *env* 2) and based on this, 41% (89/216) koalas were positive, 57% (124/216) negative, and 2% inconclusive. These genes showed higher expression in lymph node tissue from KoRV positive koalas with lymphoma compared with other KoRV positive koalas, which showed lower, fragmented expression. Terminal regions (LTRs, partial *gag*, and partial *env*) were present in SA koalas regardless of KoRV status, with almost all (99.5%, 215/216) koalas positive for *gag* 1 by proviral PCR. Further investigation is needed to understand the differences in KoRV infection in southern koala populations.

## 1. Introduction

The genome of koala retrovirus (KoRV) was sequenced in 2000 [1] and was shown to be a type-C gammaretrovirus. It is simple in structure, approximately 8.4 kb long, and contains three genes: group-specific antigen gene (*gag*), protease-polymerase gene (*pro-pol* or *pol*), and the envelope gene (*env*), flanked by long terminal repeat (LTR) regions [1]. The *gag* gene encodes matrix, capsid, and nucleocapsid proteins and is necessary for the structure [2] of KoRV and its ability to create budding virions [3]; the *pol* gene codes for reverse transcriptase, integrase and protease enzymes; and the *env* gene encodes surface and transmembrane proteins from which cell tropism and receptor affinity arise [2]. KoRV variants, denoted A-I, have been classified into three major clades, based on the hypervariable region of the *env* gene; KoRV-A, KoRV-B, and KoRV C-I [4,5,6,7,8,9,10,11]. KoRV-A is known to enter the cell via the Pit-1 (SLC20A1) receptor [12], KoRV-B via the THTR1 receptor [3,5], and the other KoRV subtypes receptors through an unknown mechanism. KoRV has two fundamental methods of transmission: endogenously through the germ line from parent to offspring [11]; and exogenously, via horizontal (close contact) or vertical (mother to joey in the pouch) transmission [5,9]. Northern mainland koala populations (from Queensland and New South Wales) have all been found to be positive for endogenous KoRV-A provirus, with exogenous variants KoRV-B through to KoRV-I also present in some koalas [8]. In contrast, southern koala populations (from Victoria and South Australia) have a lower prevalence of KoRV, primarily KoRV-A, and have lower proviral and viral loads. This has led to the theory that these populations are currently undergoing the process of KoRV endogenisation, with predominantly KoRV-A spreading exogenously through this population [8,13,14,15,16,17,18], but this is yet to be confirmed.

Studies of KoRV in South Australian (SA) koalas have been based on detection of KoRV provirus and virus using a PCR amplification of part of the *pol* gene, a highly conserved region of the KoRV genome. Based on this, large numbers of KoRV-free koalas have been found in the two main populations of SA koalas in Kangaroo Island (KI) and the Mount Lofty Ranges (MLR). However more recent studies utilising PCR targeting of *gag* and *env* genes, as well as expression studies of the entire KoRV genome, have identified a more complex KoRV profile in MLR koalas [16,17], with 38–99% positive for nine different proviral targets, and 3–51% positive for viral RNA targets [17]. It has also been shown that MLR KoRV *pol*-negative koalas produced transcripts that aligned to the KoRV genome at positions 1-1389 (5′ LTR and partial *gag* gene) and 7124-8431 (partial *env* gene and 3′LTR), with no transcription of other KoRV genes [19]. Additionally, KoRV *pol*-positive koalas had low transcription of genes between genome positions 1389 and 7124, representing the 5′ end of the *gag* gene, the *pol* gene and the 3′end of the *env* gene, unlike Queensland koalas [19]. It is currently postulated that truncated and potentially defective virus inhibits viral replication in these koalas [16,19]. Another recent study has found that terminal sequences from KoRV have, in some cases, recombined with earlier or ancient retroviral elements, such as Phascolarctid endogenous retroelement (PhER) [20,21]. PhER has been shown to still be active and expressing at least parts of genes [20,21], although the clinical significance of this is unknown. This recombination of KoRV with PhER, has been designated as recKoRV and has one prominent subpopulation designated recKoRV-1. RecKoRV-1 has been shown to be present in the MLR population, but not the KI population and notably, within the MLR population recKoRV-1 was shown to be present in KoRV *pol* negative koalas [20].

In northern koalas, disease states that have been associated with KoRV infection include lymphoid neoplasia (lymphoma and leukaemia) and chlamydiosis [15,22,23,24]. Lymphoid neoplasia and increased severity of chlamydiosis have been found to be associated with high proviral and viral loads in all regions [15,17,23] and with exogenous KoRV-B infection in northern koalas [4,5,24]. In the MLR population lymphoma has been found in koalas with KoRV viral and proviral loads comparable to those of northern koalas with neoplastic disease [17]. No significant association has been identified between KoRV-A status and *Chlamydia pecorum* status in MLR koalas [13,22], despite the exogenous transmission in this population and the suggested increased pathogenicity of exogenous KoRV infections. A study in KoRV positive koalas did show a positive correlation between KoRV viral load and severity of chlamydiosis [25], indicative of potentially increased pathogenicity with higher viral loads.

KoRV infection in SA koalas is clearly complex, and hence it is important to clarify which KoRV genes should be used to identify an individual as KoRV positive or negative, in order to understand infection prevalence and interpret disease in SA koalas. The aims of this study were to describe the integration and expression of KoRV genes in MLR koalas, so as to recommend guidelines for classification of individuals as positive or negative for infectious, full length KoRV in SA.

## 2. Materials and Methods

### 2.1. Samples

Koalas for this study (n = 216) were sourced from the Mount Lofty Ranges population in South Australia following euthanasia by veterinarians on welfare grounds. All koalas received a full necropsy at the Veterinary Diagnostic Laboratory, Roseworthy campus, University of Adelaide. Blood samples taken prior to euthanasia were stored in EDTA tubes (Vacuette^®^ Tube, Greiner Bio-One GmBH, Kremsmünster, Austria), or spleen samples were taken postmortem, and stored at −20 °C prior to DNA analysis. In koalas that were available for immediate postmortem examination (<1 h), small 0.5 cm^2^ pieces of lymph nodes were placed into RNA Later^®^ (Sigma Aldrich, Saint Louis, MO, USA), refrigerated for 24 h and stored at −80 °C prior to RNA analysis. All lymph nodes were examined histologically to determine the presence or absence of lymphoma, and if present, assigned to the lymphoma group (n = 6).

### 2.2. DNA Analysis for KoRV Provirus: DNA Extraction

DNA was extracted from whole blood or spleen for KoRV proviral analysis, using the QIAMP DNeasy Minikit (Qiagen, Hilgen, Germany) as per manufacturer’s instructions. All extracted DNA was quantified using a NanoDrop One spectrophotometer (Thermo Fisher Scientific Inc, Wilmington, DE, USA) and stored at −20 °C.

### 2.3. DNA Analysis for KoRV Provirus: DNA Proviral PCR and qPCR

KoRV proviral gene targets *gag* 1, *gag* 2, *env* 1, and *env* 2 (Figure 1) were assessed by conventional PCR using DNA extracted from either whole blood or spleen. KoRV *gag* 1 and 2 were optimised as a multiplex reaction and *env* 1 and 2 reactions were run individually. The KoRV *gag* (1 and 2) 25 μL PCR reactions consisted of 5 μL AllTaq Mastermix (Qiagen, Hilden, Germany), 0.25 μM of each *gag* 1 primer, 0.75 μM of each *gag* 2 primer (Table 1), 3 μL of 1/10 dilution DNA template, and 9 μL of PCR-grade water. The individual KoRV *env* 20 μL PCR reactions consisted of 4 μL AllTaq Mastermix (Qiagen, Hilden, Germany), 0.5 μM of each primer (Table 1), 3 μL of 1/10 dilution DNA template and 10 μL of PCR-grade water. The temperature and timing of the PCR reactions for both assays were initial activation and denaturation of 95 °C for 3 min; 40 cycles of denaturation at 95 °C for 5 s; annealing at 60 °C (KoRV *env*) or 62 °C (KoRV *gag*) for 15 s; and extension at 72 °C for 10 s. This was followed by a final extension step of 72 °C for 10 s. Gel electrophoresis using 1.0% agarose gel was carried out and PCR products visualised under UV light. Samples were loaded in combination with gel red, producing bands at the appropriate product size (Table 1).

KoRV proviral load was measured in DNA extracted from either whole blood or spleen by KoRV *pol* gene qPCR (Table 1, Figure 1). Koala β-actin was used as the reference gene. qPCR reactions of 5 µL were set up in triplicate using an Eppendorf epMotion 5075 LH model (Eppendorf, Hamburg, Germany). Each reaction consisted of 2.5 µL of Power SYBR green Master Mix (Applied Biosystems, Foster City, CA, USA), 300 nM of each primer and 1.5 µL of 1/10 or 1/100 dilution of DNA template. qPCRs were run on a 7900 HT Sequence Detection System (Applied Biosystems, Singapore). Reaction conditions were initial denaturation 95 °C for 10 min; 40 cycles of 95 °C for 15 s; 60 °C for 60 s. Melt curve analysis was carried out at 95 °C for 15 s, 60 °C for 15 s and 95 °C for 15 s. Copy number was calculated from a standard curve derived from purified PCR products for both KoRV *pol* and β-actin from koala lymphoma tissue. KoRV proviral load reported as copies/10^3^ β-actin copies.

### 2.4. RNA Extraction and Sequencing

RNA was extracted to analyse KoRV gene expression by RNAseq and qPCR. Lymph nodes (lymphomic and non-lymphomic) from eight SA koalas, two KoRV proviral positive with lymphoma, three KoRV proviral positive, and three KoRV proviral negative (except *gag* 1), were used. Tissues were thawed on ice, disrupted with zirconium oxide beads (0.5 mm, RNA-free, BioTools, Loganholme, Australia) in a Bullet Blender and Tissue Homogeniser (Next Advance, Troy, NY 12180, USA)), and then cooled on ice. The *mir*Vana™ kit (Thermo Fisher Scientific) was used to extract total mRNA as per the manufacturer’s protocol. RNA integrity (RIN) was checked using a 2100 BioAnalyser (Agilent Technologies). cDNA libraries were made and sequenced using an Illumina NovaSeq sequencing platform that generated 2 × 150 bp reads (AGRF, Melbourne, Australia).

### 2.5. KoRV Gene Expression from RNA-seq Analysis

The RNA-seq reads were cleaned using Trim_Galore (v0.4.2) [26] and AdapterRemoval (v2.2.1) [27] to remove adapters and bases with a Phred score less than 10. Fastqc (v0.11.4) [28] was used to check the quality of the cleaned reads. The reads were aligned to the koala reference genome (GCA_002099425.1_phaCin_unsw_v4.1) using HISAT2 (v2.1.0) [29].

### 2.6. BLAST Analysis

To demonstrate the presence of similar KoRV termini regions to those found in KoRV negative koalas, NCBI standard nucleotide Basic Local Alignment Search *Tool* (*BLASTn*) [30] was used to align the KoRV-A (AF151794.2) reference genome against the koala reference genome (phaCin_unsw_v4.1 reference Annotation Release 100; GCF_002099425.1). Further *BLASTn* analysis was carried out on alignments to the koala genome that only contained the terminal regions of KoRV homology. Additionally, the central “gap” sequences were recovered from between these terminal alignments from the reference genome and *BLASTn* was then used to align these gap sequences to the nucleotide NCBI database [31]. *BLASTn* was also used to align the Phascolarctos endogenous retroelement (PhER), using coordinates reported in Lober et al. 2018 [20], against the koala reference genome. PhER and KoRV alignment results were then compared to find any crossover regions.

### 2.7. Viral Two Step RT-qPCR Analysis

To quantify the difference in KoRV viral gene expression between the three groups, total mRNA was extracted from 15 koala’s lymph nodes, five KoRV proviral positive with lymphoma (Lymphoma), five KoRV proviral positive (Positive), and five KoRV proviral negative (except *gag* 1) (Negative). cDNA was reverse transcribed from 250–2500 ng of total RNA using SuperScript IV First Strand Synthesis system (Invitrogen, Thermofisher Scientific) and oligoDT primers as per manufacturer’s instructions. cDNA was diluted to a working solution based on starting RNA (equivalent to 1.25 ng/μL RNA). qPCR reactions were run, targeting four KoRV gene regions; *gag* 1, *gag* 2, *pol*, and *env* 1, with primers described in Table 1 and targets shown on the KoRV genome in Figure 1. Koala γ-actin was used as the reference gene; forward primer 5′-TGCGCAGCTTCAGATTAAACAA-3′, reverse primer 5′-GGCCTCATCACCAACATAACTG-3′, due to increased stability in comparison to koala ß-actin across RNA-seq data for the cohorts. Coefficients of variance (CoV) [32] were calculated from gene expression normalised counts (cpm) across all samples with γ-actin CoV calculated as 0.04 (position 289/14435) in comparison to ß-actin CoV 0.06 (position 906/14435). qPCR was optimised for γ-actin and conditions for the qPCR are as follows; 5 µL reactions were set up in triplicate using an Eppendorf epMotion 5075 LH model (Eppendorf, Hamburg, Germany). Each reaction consisted of 2.5 µL Power SYBR green Master Mix (Applied Biosystems, Foster City, CA, USA). 300 nM of each primer (Table 1) and 1.5 µL of cDNA template. qPCRs were run on a 7900 HT Sequence Detection System (Applied Biosystems, Singapore). Reactions conditions were initial denaturation 95 °C for 10 min, followed by 40 cycles of 95 °C for 15 s, 60 °C for 60 s. Melt curve analysis was carried out with conditions of 95 °C for 15 s, 60 °C for 15 s, and 95 °C for 15 s.

### 2.8. Relative qPCR Statistical Analysis

Comparative CT (2^(−∆∆Ct)^) was calculated for each of the four gene targets studied. The Mann-Whitney U Test was used to determine significant differences between groups. Both calculations were carried out in Microsoft Excel (Version 16.42, 2020).

## 3. Results

### 3.1. Proviral DNA Analyses

The presence/absence of three KoRV genes was determined by PCR using five targets, two in the *gag* gene, one in the *pol* gene and two in the *env* gene, the targets called *gag* 1, *gag* 2, *pol*, *env* 1, and *env* 2, respectively. The majority of koalas (212/216) could be placed into two categories: positive for all gene targets or negative for all gene targets except *gag* 1 (Table 2). The remaining koalas (4/216) showed variation in their gene target results with one koala negative for all targets (Table 2).

### 3.2. KoRV Gene Expression from RNA-seq Analysis

Figure 2 shows the KoRV expression profiles for the lymph nodes of eight koalas as determined by mapping the RNA-seq reads to a KoRV insertion found in the reference koala genome (scaffold 27:2807277–2815707 of GCA_002099425.1_phaCin_unsw_v4.1). Every RNAseq sample, regardless of lymphoma and KoRV PCR status, showed substantial coverage across the LTRs and nearby gene regions (herein called the terminal regions).

Two positive koalas with lymphoma (Lymphoma 1 and 2) also showed substantial, comparable coverage across the region between them (herein called the central region), supporting previous findings of higher KoRV transcription and viral loads in lymphoid neoplasia [23]. Three koalas negative for lymphoma but PCR positive for all five proviral gene targets (Positive 1, 2 and 3) only showed only low, fragmented coverage across the central region. Low loads of KoRV are likely due to only exogenous infections assumed in South Australia [10], and possibly subclinical retroviral status. Of three koalas negative for all proviral PCR targets except *gag* 1, the koalas Negative 1 and 2 showed no coverage across the central region, but Negative 3 showed low, fragmented coverage. No transcription and no provirus found in the central region would deem these koalas unable to produce virions and support their KoRV negative infections status. Negative 3 koala could be an acute or early-stage infection, potentially due to the target tissue of lymph nodes from which the RNA was extracted for RNAseq, whereas proviral analysis was carried out on blood or spleen tissue; alternatively, this may reflect a KoRV positive koala whose proviral load is too low for PCR positivity.

### 3.3. Terminal Regions of Homology to KoRV

Alignment of KoRV-A (AF151794.2) against the reference koala genome (phaCin_unsw_v4.1 reference Annotation Release 100; GCF_002099425.1) using BLASTn showed two types of alignments, a smaller subset of those found in Hobbs et al. [33]: nine alignments were to the whole or most of KoRV, but eight only aligned to the terminal regions of KoRV. The terminal alignments were found in scaffolds, 137, 005, 288, 113, 073, 357, 253, and 101 (coordinates in Appendix A). The terminal region alignment lengths varied from 1155–2663 bp in the 5′ region and 812-2919 bp in the 3′ region and are quite similar to the consistently expressed terminal regions in Figure 2. The central regions within these scaffolds, from the reference genome, were then aligned using BLASTn against the nucleotide database. Alignments were found to uncharacterised loci in the koala, wombat (*Vombatus ursinus*) and gray short-tailed opossum (*Monodelphis domestica*) genomes. However, alignment of PhER (phaCin_unsw_v4.1.fa.scaf00062: 10912078-10920108, as per Lober et al. 2018 [20]) using BLASTn against the koala reference genome revealed alignment to the same scaffolds and positions of the central gap region of KoRV (coordinates in Appendix A). Overall, the BLASTn data shows that similar terminal regions are present in the northern koala reference genome and the “gap” region, in the identified scaffolds, aligns to PhER (Figure 3), reiterating findings in Lober et al. [20] and Hobbs et al. [33]. The significant similarity of terminal fragments in the reference genome and within the KoRV negative koalas in SA is suggestive of the presence of a recombinant KoRV (recKoRV) in these koalas.

### 3.4. KoRV Gene Expression from Viral Two Step RT-qPCR Analysis

qPCR for KoRV gene targets *gag* 1, *gag* 2, *pol*, and *env* 1 were carried out on RNA from fifteen koala lymph nodes, five KoRV positive with lymphoma (three of which were from a previous study cohort [25]), five KoRV positive, and five KoRV negative koalas. Seven of the eight koalas that were previously examined by RNA-seq transcriptome analysis were examined by viral RT-qPCR (all except Negative 3, as not enough material remained). Because *gag* 1 was positive in almost every test, regardless of the sample lymphoma and KoRV status, it was excluded from further analysis. Ten KoRV proviral gene (DNA) positive koalas, including five with lymphoma, were also positive for the three KoRV viral gene (RNA) targets. Five koalas that were KoRV proviral gene (DNA) negative were also negative for the three KoRV viral gene (RNA) targets. Table 3 shows the comparative CT mean fold change (2^−∆∆CT^) and FC range (2^−(∆∆CT±sd)^) between KoRV positive and lymphomic koalas (expanded results Appendix A). Gene targets *pol* and *env* 1 had significantly higher expression for KoRV proviral and viral positive koalas with lymphoma versus those that were KoRV proviral and viral positive.

The targeted viral expression reiterated the expression demonstrated with RNAseq data, with the highest expression of KoRV genes in koalas with lymphoma. The koalas tested on both proviral (DNA) qPCR and viral (RNA) qPCR analyses demonstrated consistency in results, increasing confidence in these assays.

## 4. Discussion

The findings of this study demonstrate the complexity of KoRV status in SA koalas, but clearly show that the central region of KoRV *(gag* 2, *pol*, *env* 1, and *env* 2) is diagnostic for classifying individuals as KoRV positive (infected) or negative (non-infected). The highest expression of all KoRV genes was by koalas with lymphoma, whereas non-lymphomic KoRV positive koalas had variable expression profiles in the central region of the KoRV viral genome. The koalas tested on both proviral (DNA) qPCR and viral (RNA) qPCR analyses demonstrated consistency in results. Gene target *gag* 1 was positive in almost all MLR koalas (215/216) regardless of KoRV status and the terminal sequences inclusive of this are present numerous times in the koala reference genome.

Based on the central region of KoRV, proviral PCR revealed two distinct groupings: 41.2% (89/216) positive and 57.4% (124/216) negative for all targets. These results differ to those of a recent study in which MLR koalas were shown to have varying prevalence of each of the KoRV genes within their proviral DNA, with 79% (77/97) PCR positive for at least one target on each proviral genes, 99% (96/97) positive for *pol* gene target and only 41% (40/97) positive for one of the *env* gene targets [17]. Viral expression (qPCR) supported our proviral analyses since all KoRV negative koalas showed no expression of *gag* 2, *pol*, and *env* 1 targets. However, in this study there were a low number (1.4%, 3/216) of koalas that showed mixed results in the proviral PCRs and this could be due to the sensitivity of these assays, as KoRV proviral load can be very low in SA koalas, or potentially represent acute infections.

Gene expression in KoRV positive koalas showed variable depth of coverage across the central region, except for lymphomic koalas in which it was consistently high. High expression of KoRV was shown directly from lymphomic tissue infected with exogenous KoRV and supports the ongoing assumption that oncogenesis occurs is associated with retroviral infection in koalas [22,23,34,35]. However, further studies to like causation beyond observation of upregulation of expression clonal transformed cells are needed and are the subject of future investigations with common KoRV insertion sites observed near oncogenes in koalas with cancer [36]. From RNAseq alignment to KoRV-A genome, it can be seen that there is more variance in the *env* region in the lymphoma samples than the KoRV positive samples (Figure 2), further investigations of the KoRV variants in this cohort could be undertaken to determine the role of different subtypes in oncogenesis, as found in other studies [5,37]. Suppression, latency, or subclinical infections of KoRV may explain the fragmentation of alignment to the central region of KoRV in KoRV positive, non-lymphomic koalas. In order for the host to survive, several anti-viral strategies have evolved in response to retroviral infections. Small piwi interacting RNA (piRNA) molecules or small interacting RNA (siRNA) have been found to silence transposons in germinal cells [38,39,40], DNA methylation can also inhibit transcription and several host genes have been shown to have antiretroviral properties [41]. piRNAs have been found in koalas, highly expressed within testicular tissue, but also expressed in other tissues including brain, liver, and lymph nodes [21]. The inhibition shown from piRNAs through ping-pong modelling in koalas, corresponds to central regions of the KoRV genome [21]. Therefore, piRNAs could be responsible for some of the inhibition and fragmentation of the transcription of central genes in SA koalas as seen in KoRV positive koalas. The low gene expression found in SA koalas differs to that found in previous studies of endogenised Queensland koalas, in which KoRV expression and viral loads are high [17,23,25,42]. Cellular genes that encode restriction factors, such as those belonging to the cytidine deaminases, have shown significant antiretroviral properties, including inhibition of viral transcription. Examples of antiretroviral cellular genes are *APOBEC3*, *SAMHD1*, and *MX2* [41]. A recent study found the addition of a mouse homolog of *APOBEC3* significantly reduces the infectivity of KoRV in vitro [43]. The koala homolog to human or mouse *APOBEC3* gene is not present, but *APOBEC1*, *APOBEC2 APOBEC4*, and *AICDA* genes (other cytidine deaminases) are annotated on the koala genome (NCBI database), with their effect yet to be demonstrated on KoRV infectivity.

From the RNAseq data, in two of the three KoRV proviral (DNA) negative koalas there was no detectable expression from the central region, but the other one koala showed a similar pattern of expression to the KoRV proviral (DNA) positive koalas. Unfortunately, this particular koala did not undergo viral (RNA) gene expression analysis through qPCR. The proviral PCRs were carried with blood or spleen and RNA-seq with lymph node tissue, this difference could support lymph nodes as the primary target for infection and replication. Therefore, our tentative conclusion is that this koala was recently infected with the virus only present in the lymph nodes and had not yet spread to other tissues including the blood and spleen. For those KoRV negative koalas without proviral DNA detection or proviral gene expression in the central region of the KoRV genome it would be highly unlikely that these koalas would be able to produce infectious KoRV virions. *Env* genes are responsible for entry into the cells, coding for the receptor binding domain and the transmembrane unit, while *pol* genes are responsible for enzymes including viral polymerase, *gag* genes code for structural proteins and allow for formation of a virion [44,45]. Therefore, without these elements, formation of KoRV virions would be impossible, but it would be necessary to carry out in vitro infection studies in cell culture to confirm this.

There were multiple sequences in the koala reference genome that showed strong homology to the KoRV terminal regions but not the central region. Some of these sequences have the KoRV central region replaced by sequences from the central region of PhER. It is possible that these insertions in the reference genome have been generated by recombination between the termini of PhER and KoRV, attributed to recKoRV in previous studies [20,33]. We note that complete KoRV and PhER retroviral sequences only show 72% identity over 102 nucleotides around the 7500 bp region on the KoRV genome, which may be enough for recombination to have occurred. Additionally, the lack of KoRV in significant numbers of probable recKoRV positive koalas elicits evolutionary questions for this virus, unable to be answered without greater investigations. A possible alternative explanation for these terminal regions could be due to the historical foundation of this population, reintroduced to South Australia from very small numbers of Victorian koalas. Due to the genetic bottleneck, partial gene loss may have occurred in this population, with endogenisation of a defective version of KoRV in SA koalas. ERVs that have lost the *env* gene tend to have increased proliferation within the host genome, with less defined loci of integration [46], therefore the *env*-less terminal regions within this study could have increased proliferation, and in turn, increase transcription of KoRV fragments. Further long read genome sequencing of KoRV positive and negative koalas could help answer some of the questions surrounding these terminal regions.

There is the potential that the terminal retroviral expression in KoRV negative koalas could be inhibiting or reducing subsequent exogenous spread of KoRV in SA populations. These elements could also be reducing loads in KoRV positive koalas. Protective antiretroviral properties have been encountered in the study of endogenous retroviruses (ERVs). Some endogenous retroviral elements can inhibit other retroviral entry into cells and therefore increase the resistance to future retroviral infection, such as in feline leukemia virus (FeLV) and murine leukemia virus (MLV) [45,47,48,49]. *Env* regions have been shown to be recruited in response to exogenous retrovirus (XRV) infection to block viral receptors [50] and has been experimentally demonstrated in mice [49], cats [47], and chickens [51]. The transcribed KoRV fragments from the 5′ LTR and 5′ *gag* region and from the 3′ *env* and 3′ LTR could be part of the resistance to future KoRV infection, as suggested by Tarlinton et al. 2017 [19]. This would assume that these elements are endogenised throughout the population, which is likely since the RNA-seq alignment mapping showed presence of these regions in all koalas and proviral PCR analysis showed an overwhelming majority (99.5%, 215/216) of koalas were positive for the 3′ *gag* gene target (*gag* 1). In vitro studies could explore the ability for KoRV negative lymphocytes to withstand KoRV infection and the effect of transcripts from 5′ and 3′ regions of KoRV on infection rates. These studies would help distinguish clinical relevance of these KoRV transcripts and expand our understanding of KoRV transmission and cellular response.

## 5. Conclusions

South Australian koala populations are intriguing in their KoRV and other retroviral integration and expression patterns, differentiating them from their more extensively studied northern counterparts. The clinical relevance of KoRV in SA requires the definition between KoRV positive and KoRV negative koalas and this large cohort study demonstrated two clear groupings, allowing for future delineation. The commonly used *pol* qPCR should detect positive cases, however confirmation with *gag* 2 and *env* 1 gene targets would increase confidence, highlighting this central region as the most valuable for differentiation. This study also highlighted the homology to the terminal regions of KoRV in all koalas. Sequencing the termini and gap regions, or potentially long read genome sequencing, of KoRV negative SA koalas could increase understanding of what these regions are and how they are situated in the genome. In vitro cell culture studies using KoRV negative koalas could show whether they have a level of resistance to KoRV infection. KoRV negative koalas in SA are important for understanding not only infection and transmission, but also evolution of KoRV.

## Figures and Tables

**Figure 1 animals-11-01477-f001:**
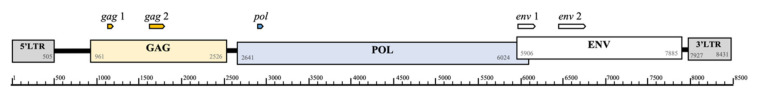
Representation of the KoRV genome (AF151794); showing genes, coordinates and primer locations with block arrows indicating amplicon length.

**Figure 2 animals-11-01477-f002:**
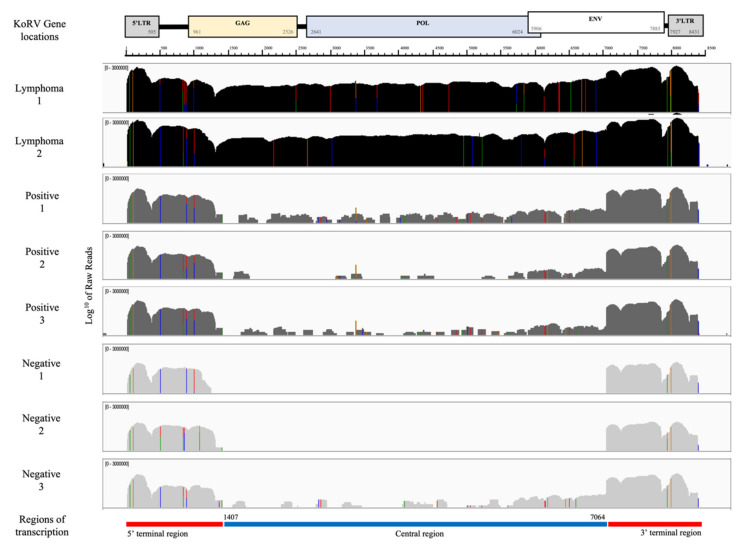
Transcription expression profile (mapped cleaned reads) for two koalas with lymphoma, three KoRV proviral positive and three KoRV proviral negative (except *gag* 1) koalas, demonstrating alignment to the KoRV genome, location of primers for proviral PCRs and transcription regions. All expression is a log_10_ scale of read counts, black represents KoRV proviral positive koalas diagnosed with lymphoma, dark grey represents KoRV proviral positive koalas and light grey represents KoRV proviral negative (except *gag* 1) koalas. All mapped to scaffold 27; 2807277–2815707 (GCA 002099425.1 phaCin_unsw_v4.1), KoRV-A accession AF151794.

**Figure 3 animals-11-01477-f003:**
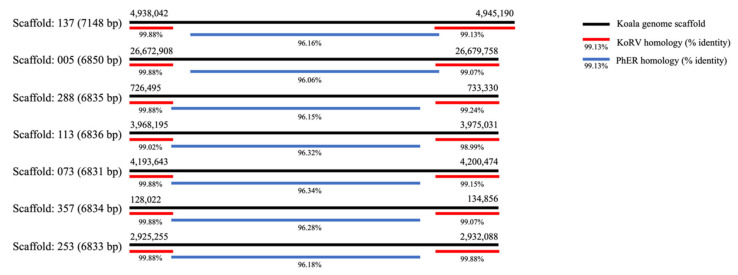
BLASTn analysis results showing crossover of the central gap regions from alignment of KoRV termini regions (red) and PhER (blue) to the koala reference genome (black). KoRV is represented by the reference sequence for KoRV-A: AF151794, PhER: phaCin_unsw_v4.1. fa.scaf00062:10912078–10920108, as per Lober et al. 2018 [20] and the reference koala genome: phaCin_unsw_v4.1 reference Annotation Release 100; GCF_002099425.1.

**Table 1 animals-11-01477-t001:** PCR targets and primers.

Target	Primers	Product Size	Reference
Koala ß-actin	Fwd 5′-GAGACCTTCAACACCCCAGC-3′ Rev 5′-GTGGGTCACACCATCACCAG-3′	111 bp	Shojima et al. 2013 [4]
KoRV *gag* 1	Fwd 5′-CGGACCAGGTTCCCTACATC-3′ Rev 5′- TCGCCCGTTATCTTGACCAG-3′	110 bp	This study
KoRV *gag* 2	Fwd 5′-TTGGCCTTTCTCCTCAGCAG-3′ Rev 5′-CCGTGTTGTGATCCCACTGA-3′	290 bp	This study
KoRV *pol*	Fwd 5′-TTGGAGGAGGAATACCGATTACAC-3′ Rev 5′-GCCAGTCCCATACCTGCCTT-3′	111 bp	Tarlinton et al. 2005 [23]
KoRV-A *env* 1	Fwd 5′-TCCTGGGAACTGGAAAAGAC-3′ Rev 5′-GGGTTCCCCAAGTGATCTG-3′	321 bp	Waugh et al. 2017 [24]
KoRV *env* 2	Fwd 5′-GCCCTCGGCCCTCCTTATTA-3′ Rev 5′-GGCAATCTGGAGGCTAGTCAA-3′	522 bp	Sarker et al. 2020 [17]

**Table 2 animals-11-01477-t002:** Proviral PCR results.

KoRV Gene Target					Count (n = 216)	%	Median (Range) of Proviral Load KoRV Copies/ 10^3^ ß-Actin Copies *
*gag 1*	*gag 2*	*pol*	*env 1*	*env 2*			
+	+	+	+	+	89	41.2	41 (0.2–270)
+	-	-	-	-	123	56.9	0
+	-	-	-	+	2	0.9	0
+	-	-	+	-	1	0.5	0
**-**	-	-	-	-	1	0.5	0

* calculated from *pol* gene qPCR.

**Table 3 animals-11-01477-t003:** Comparative CT method results (fold change 2^−∆∆CT^) of KoRV central genes from for lymph node RNA from KoRV positive koalas with lymphoma relative to KoRV positive koalas.

KoRV Gene	2^−∆∆CT^ Lymphomic Koalas Relative to Positive Koalas		
	Mean FC *	FC Range	*p* Value
*gag* 2	299	47.7–1880.5	0.060
*pol*	**602**	185.8–1950.2	0.012
*env* 1	**242**	42.5–1375.6	0.012

FC = Fold change calculated by 2^−∆∆CT^. * Bold values are significant (Mann-Whitney U test, *p* < 0.05), more detail of calculations and samples in Appendix A.

## Data Availability

The data presented in this study are available on request from the corresponding author. The data are not publicly available due to ongoing research commitments.

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
