# Peer review of "Molecular Diagnosis of Koala Retrovirus (KoRV) in South Australian Koalas (Phascolarctos cinereus)"

_animals, 2021, doi:10.3390/ani11051477_

Round 1

Reviewer 1 Report

The authors describe several approaches to characterize the presence and expression of five fragments representing the Koala Retrovirus genome with the intention of providing a reliable method of differentiating infected and non-infected individuals in future research.   Overall the study appears to be well conceived and the results are valuable. However, I feel that the presentation would benefit from greater clarity in the nomenclature used to describe the different subgroups of koalas. It would also be useful to the uninitiated reader to state the objective of each of the individual methods used as this is not done in all in cases (e.g. not clearly stated for “DNA proviral PCR and qPCR” or “RNA extraction and sequencing”, whereas the rationale is clearly stated in the first sentence of the section describing the “Viral two step RT-qPCR analysis” methods)   A few specific comments:  

Line 188: “To quantify the difference in KoRV viral gene expression between the three groups” …unclear what groups are being referred to. Presumably your three groups are 'proviral positive, lymphoma positive’, 'proviral positive, lymphoma negative’, and 'proviral negative, lymphoma negative’. If so, this could be worded more clearly (perhaps using the shorthand nomenclature used in Figure 2). It is also unclear what criteria were used for designating a koala into proviral positive or proviral negative groups (were all gene fragments used? …or only some?).

  Lines 242-3: “Low loads of KoRV are likely due to exogenous infection, and possibly subclinical retroviral status”. It is unclear what the basis for this unreferenced statement is. It might be better to remove this from the results and expand on it in the discussion as it is unclear why endogenous cases can be excluded here.   Lines 247-9: “Negative 3 koala could be an acute infection due to the target tissue of lymph nodes…”. Unclear what you mean by an “acute infection” (early stage infection perhaps?). Also probably more appropriate to be discussed in more detail in the discussion section as this koala could be considered a mis-categorisation of a “positive” koala.   Line 304: Would it not be more appropriate to classify koalas as being “infected” and “non-infected”? …as “infectious” and “non-infectious” implies an ability to transmit the virus. Presumably endogenous transmission is likely (inevitable?) in all reproductively active infected animals, but is an infected animal always capable of exogenous infection?   Lines 311-322: It is unclear why the results of the current study are different to the results from the previous study when the same population is being studied. Can the authors clarify, as this is central to their conclusion that these are valuable markers for differentiating positive and negative koalas?   Line 322: Sentence incomplete.

Author Response

Please see attachment, thank you.

Reviewer 2 Report

Molecular diagnosis of koala retrovirus (KoRV) in South Australian koalas (Phascolarctos cinereus)

The manuscript my Stephenson et al. investigates the presence and expression of the koala retrovirus (KoRV) in southern koala populations.

The prevalence of KoRV is known to differ between the koala populations of northern and southern Australia. In the northern koala population, all individuals are positive for endogenous KoRV-A, which is a recent and ongoing endogenisation process with multiple negative health effects. Exogenous variants of KoRV are also present in the northern population.  Although it is known that there are far fewer KoRV positive koalas in southern Australia, the specifics of KoRV infection in this population remain unclear.  Investigating KoRV in southern koalas is of interest to analyse the effects of KoRV on koala health (in comparison to northern populations) and from the perspective of the spread and evolution of a retrovirus during the very early stages of retroviral endogenisation.

Here the authors use PCR to detect the presence of various provial genes in southern koalas. Around 40% of koalas are positive for the five PCR tests across the gag, pol and env genes.  However, in the koalas that are negative for the pol and env genes, all but one test positive for the first gag PCR; they appear to generally lack most of the proviral sequence, except for the terminal regions.  This suggests that there is a mutated version of KoRV in the southern population. 

A pre-print on the bioRxiv preprint server (R.E Tarlinton et al. Differential and defective expression of Koala Retrovirus reveal complexity of host and virus evolution. bioRxiv 211466; reference 19 in this manuscript) contains similar information about the presence of a mutated KoRV in southern populations, where only the terminal regions of the virus are present. Tarlinton et. al. also show some expression data (RNAseq) of the proviral genes from northern and southern koalas. 

Here, the authors extend the Tarlinton et. al. study by including a much larger number of individuals (216 compared to 19 southern koalas), although only a small subset are measured for proviral expression. The data also includes koalas suffering from lymphomas. 

Overall, it is a nice study of a large population of koalas, with PCR, qPCR and RNAseq data all leading to the same conclusion.  However, I think the results could be presented more clearly (details below). Also, the authors present RNAseq data but do not analyse it in much detail and some further information from that data could be useful. 

Comments:

Line 110: “Samples were all deemed unfit for release due to disease or trauma”.  What types of disease did these koalas have and are they related to KoRV load?  Could the number of koalas with full length KoRV proviruses be skewed by the fact that the sample is not random, but contains a lot of sick koalas? In particular, how many koalas had lymphoma and how were these cases distributed between the proviral PCR groups?  Is there any association between presence of full length KoRVs and number of lymphoma cases? 

Table 3.  It would be much better to display boxplots of the qPCR expression data (e.g. in the style of Figure 3 from the paper by Tarlinton et. al [reference 19 here]). For example, the –ΔCT values for each sample for each separate region of KoRV could be displayed, which would give a better overview of the variance in each category.  When the range is from 47 – 1880 then giving the mean is not very informative, especially when the difference in expression is not significant.  A boxplot figure (for all three categories) could then be displayed in a figure panel together with Figure 2 because it would be useful to see the RNAseq expression profile of mapped reads and the qPCR data together. 

In the related Table S1, the fold differences are slightly nonsensical if one category has no expression because this actually makes the fold difference infinity.  As such, the massive fold changes don’t really give any indication into the level of expression in the KoRV group.  Even if the expression is low, the fold change will be massive when compared to a value of zero or close to zero.  As far as I know, qPCR CT values above 35 should not be used for calculations.  Was a “no template” control used for the qPCRs?  If so, values above the no template control definitely should not be used. 

Lines 240 and 325: “…supporting the role of KoRV in oncogenesis” and “…supports the ongoing assumption that retroviral oncogenesis occurs in koalas”.

This is a bit too vague and should probably be re-worded, e.g. “...which is in line with previous studies showing that high KoRV loads are associated with lymphomas”.  Also, since they find interesting expression relating to lymphomas then maybe it would be good to take a few sentences to try to explain these results further.  In general, endogenous retroviruses can become highly activated after cellular transformation due to a decrease in cellular constraints (hypomethylation). This happens even in cases where the cancer may not be caused by retroviral oncogenesis (e.g. high level of ERV expression in human cancers). Although it is likely that retroviral oncogenesis can occur via KoRV (See paper by McEwen et al. 2021, Nat Commun 12:1316), high levels of KoRV alone do not support the role of KoRV in oncogenesis. In lymphoma samples, the high level of full length KoRV could well be due to (i) a decrease in cellular constraints that normally keep the expression of retroviruses at a low level and/or (ii) the clonal expansion of cells that contain exogenous KoRV variants which have triggered transformation by insertional mutagenesis. In both cases, KoRV load increases only after cells are transformed.

It would also be interesting to see a slightly more in depth analysis of the RNAseq data.  For example, it is not clear if these KoRVs are KoRV-A or other KoRV variants which could be interesting to find out (e.g. by looking at the hyper-variant envelope region from bases 6185 -6319 in the KoRV-A genome).  In Tarlinton et al. (19) they find KoRV-A, KoRV-E and KoRV-I.  Also, from Figure 2, it seems like the three positive koalas have identical SNPs across the whole aligned region whereas the two lymphoma cases have different SNPs (from each other and from the positive variants). It is possible that certain KoRVs subtypes are more likely to cause lymphomas and so it could be possible that screening koalas by KoRV subtype will be needed in the future during molecular diagnosis.  This is far beyond the scope of this study but could be mentioned.

Line 250. 3.3. Terminal regions of homology to KoRV

The presence of recKoRV in the published koala genome is already well documented.  The koala from which the reference genome was sequenced (Bilbo) has over100 KoRV/KoRV derived integrations. (Hobbs et al. 2017, Sci Rep 7, 15838).    Only a small number of these are actually in the reference genome due to difficultly with assembly (I think only KoRVs/recKoRVs spanned by single PacBio reads were included in the release).   The Supplementary Table in Hobbs et al. lists all of the identified KoRV and recKoRV integrations and their genomic locations in the reference genome. This makes the Blast carried out here somewhat redundant with the recKoRVs identified being only a subset of the total. Section 3.3 and also legends of Tables S3 and S4 should be revised to reflect the true number of KoRVs/recKoRVs found in northern koalas. 

Line 264: “Overall, the BLAST data shows that the terminal regions are present in several copies in the koala reference genome”.  As above, the reference genome is derived from a northern koala, but the assembly is not complete and does not contain the full set of KoRVs or recKoRVs present in the sequenced koala so this statement is somewhat misleading.   

Line 268: “The significant similarity of terminal fragments in the reference genome and within the KoRV negative koalas in SA is suggestive of the presence of a recombinant KoRV (recKoRV) in these koalas”.  Why is no data presented on RNAseq reads that align to recKoRVs (the central PhER region) in the koala genome? Since they have already aligned all the reads to the koala genome it would not be hard to do this.  It would be very interesting to know if the mutated KoRVs in MLR koalas contain a similar type of recKoRV (containing PhER) to northern koalas. The presence of recKoRV would be very interesting from both an evolutionary and a biological point of view and may explain the pattern of coverage from the RNAseq data.

Line 378: “ERVs that have lost the env gene tend to have increased proliferation within the host genome, with less defined loci of integration[44], therefore correspond to the increased terminal reads seen in this study”. The study referenced here (Magiorkinis et al. 2012) is based on ancient retroviral endogenisations (in the region of 25 - 100 million years ago), whereas KoRV endogenisation occurred > 50,000 years ago.  I’m not sure the same evolutionary principles apply at such an early stage.  In northern koalas, intact KoRVs show high proliferation in the population with no defined (fixed) loci of integration (McEwen et al. 2021, Nat Commun 12:1316). Over millions of years, it is highly likely that env-less KoRVs will be found at a much higher proportion in the genome than more intact KoRVs, but at this early stage it is still puzzling that southern koalas show such a different pattern of KoRVs/env-less KoRVs to northern koalas.  Since the evolutionary mechanisms may be different from the one referenced, so it might be better for the authors to be less definitive in their statement. 

Line 383: “There is the potential that the terminal retroviral expression in KoRV negative koalas could be inhibiting or reducing subsequent spread of KoRV in SA populations”.  Is there any reason why northern koalas show such a different outcome, despite also containing KoRVs that are missing the central region? 

Other minor comments:

Line 26:decreased prevalence in southern populations”.  The authors use the terms “decreased” or “increased” when they mean “lower” or “higher”. In a scientific context “decreased” means “to become smaller” and would normally be used in gene expression analysis where samples are measured before and after treatment, for example.  Would be better to write something like “...has lower prevalence in southern populations, with correspondingly lower proviral and viral loads...”. 

The same issue is found in lines 286 and 290 where “increased” should be “higher”

Line 27/28: “KoRV has been associated with lymphoid neoplasia, i.e. lymphoma”.  Here “i.e.” should be “e.g.” because lymphoid neoplasia can also be lymphocytic leukemias etc. 

Line 159: Please mention the specific tissues that were sequenced, and also on line 189.  Will RNA-seq data be uploaded to a public respositoty?

Line 343: Genes are normally presented in italics to distinguish them from proteins.

Author Response

Please see attachment, thank you.
